# ESCHER: Eschewing Importance Sampling in Games by Computing a History Value Function to Estimate Regret

**Stephen McAleer**
Carnegie Mellon University
smcaleer@cs.cmu.edu

**Gabriele Farina**
Carnegie Mellon University
gfarina@cs.cmu.edu

**Marc Lanctot**
DeepMind
lanctot@deepmind.com

**Tuomas Sandholm**
Carnegie Mellon University
Strategy Robot, Inc.
Optimized Markets, Inc.
Strategic Machine, Inc.
sandholm@cs.cmu.edu

## Abstract

Recent techniques for approximating Nash equilibria in very large games leverage neural networks to learn approximately optimal policies (strategies). One promising line of research uses neural networks to approximate counterfactual regret minimization (CFR) or its modern variants. DREAM, the only current CFR-based neural method that is model free and therefore scalable to very large games, trains a neural network on an estimated regret target that can have extremely high variance due to an importance sampling term inherited from Monte Carlo CFR (MCCFR). In this paper we propose an unbiased model-free method that does not require any importance sampling. Our method, ESCHER, is principled and is guaranteed to converge to an approximate Nash equilibrium with high probability. We show that the variance of the estimated regret of ESCHER is orders of magnitude lower than DREAM and other baselines. We then show that ESCHER outperforms the prior state of the art—DREAM and neural fictitious self play (NFSP)—on a number of games and the difference becomes dramatic as game size increases. In the very large game of dark chess, ESCHER is able to beat DREAM and NFSP in a head-to-head competition over $90\%$ of the time.

## 1 Introduction

A core challenge in computational game theory is the problem of learning strategies that approximate Nash equilibrium in very large imperfect-information games such as Starcraft (Vinyals et al., 2019), dark chess (Zhang & Sandholm, 2021), and Stratego (McAleer et al., 2020; Perolat et al., 2022). Due to the size of these games, tabular game-solving algorithms such as *counterfactual regret minimization (CFR)* are unable to produce such equilibrium strategies. To sidestep the issue, in the past stochastic methods such as *Monte-Carlo CFR (MCCFR)* have been proposed. These methods use computationally inexpensive unbiased estimators of the regret (i.e., utility gradient) of each player, trading off speed for convergence guarantees that hold with high probability rather than in the worst case. Several unbiased estimation techniques of utility gradients are known. Some, such as *external sampling*, produce low-variance gradient estimates that are dense, and therefore are prohibitive in the settings mentioned above. Others, such as *outcome sampling*, produce high-variance estimates that are sparse and can be computed given only the realization of play, and are therefore more appropriate for massive games.

However, even outcome-sampling MCCFR is inapplicable in practice. First, since it is a tabular method, it can only update regret on information sets that it has seen during training. In very large games, only a small fraction of all information sets will be seen during training. Therefore,

generalization (via neural networks) is necessary. Second, to achieve unbiasedness of the utility gradient estimates, outcome-sampling MCCFR uses importance sampling (specifically, it divides the utility of each terminal state by a reach probability, which is often tiny), leading to estimates with extremely large magnitudes and high variance. This drawback is especially problematic when MCCFR is implemented using function approximation, as the high variance of the updates can cause instability of the neural network training.

Deep CFR (Brown et al., 2019) addresses the first shortcoming above by training a neural network to estimate the regrets cumulated by outcome-sampling MCCFR, but is vulnerable to the second shortcoming, causing the neural network training procedure to be unstable. DREAM (Steinberger et al., 2020) improves on Deep CFR by *partially* addressing the second shortcoming by using a history-based value function as a baseline (Schmid et al., 2019). This baseline greatly reduces the variance in the updates and is shown to have better performance than simply regressing on the MCCFR updates. However, DREAM still uses importance sampling to remain unbiased. So, while DREAM was shown to work in small artificial poker variants, it is still vulnerable to the high variance of the estimated counterfactual regret and indeed we demonstrate that in games with long horizons and/or large action spaces, this importance sampling term causes DREAM to fail.

In this paper, we introduce *Eschewing importance Sampling by Computing a History value function to Estimate Regret (ESCHER)*, a method that is unbiased, low variance, and does not use importance sampling. ESCHER is different from DREAM in two important ways, both of which we show are critical to achieving good performance. First, instead of using a history-dependent value function as a baseline, ESCHER uses one directly as an estimator of the counterfactual value. Second, ESCHER does not multiply estimated counterfactual values by an importance-weighted reach term. To remove the need to weight by the reach to the current information state, ESCHER samples actions from a fixed sampling policy that does not change from one iteration to the next. Since this distribution is static, our fixed sampling policy simply weights certain information sets more than others. When the fixed sampling policy is close to the balanced policy (i.e., one where each leaf is reached with equal probability), these weighting terms minimally affect overall convergence of ESCHER with high probability.

We find that ESCHER has orders of magnitude lower variance of its estimated regret. In experiments with a deep learning version of ESCHER on the large games of phantom tic tac toe, dark hex, and dark chess, we find that ESCHER outperforms NFSP and DREAM, and that the performance difference increases to be dramatic as the size of the game increases. Finally, we show through ablations that both differences between ESCHER and DREAM (removing the bootstrapped baseline and removing importance sampling) are necessary in order to get low variance and good performance on large games.

## 2 BACKGROUND

We consider extensive-form games with perfect recall (Osborne & Rubinstein, 1994; Hansen et al., 2004; Kovařík et al., 2022). An extensive-form game progresses through a sequence of player actions, and has a *world state* $w \in \mathcal{W}$ at each step. In an $N$-player game, $\mathcal{A} = \mathcal{A}_1 \times \cdots \times \mathcal{A}_N$ is the space of joint actions for the players. $\mathcal{A}_i(w) \subseteq \mathcal{A}_i$ denotes the set of legal actions for player $i \in \mathcal{N} = \{1, \ldots, N\}$ at world state $w$ and $a = (a_1, \ldots, a_N) \in \mathcal{A}$ denotes a joint action. At each world state, after the players choose a joint action, a transition function $\mathcal{T}(w, a) \in \Delta^{\mathcal{W}}$ determines the probability distribution of the next world state $w'$. Upon transition from world state $w$ to $w'$ via joint action $a$, player $i$ makes an *observation* $o_i = \mathcal{O}_i(w')$. In each world state $w$, player $i$ receives a utility $u_i(w)$. The game ends when the players reach a terminal world state. In this paper, we consider games that are guaranteed to end in a finite number of actions and that have zero utility at non-terminal world states.

A *history* is a sequence of actions and world states, denoted $h = (w^0, a^0, w^1, a^1, \ldots, w^t)$, where $w^0$ is the known initial world state of the game. $\mathcal{U}_i(h)$ and $\mathcal{A}_i(h)$ are, respectively, the utility and set of legal actions for player $i$ in the last world state of a history $h$. An *information set* for player $i$, denoted by $s_i$, is a sequence of that player's observations and actions up until that time $s_i(h) = (a_i^0, o_i^1, a_i^1, \ldots, o_i^t)$. Define the set of all information sets for player $i$ to be $\mathcal{I}_i$. The set of histories that correspond to an information set $s_i$ is denoted $\mathcal{H}(s_i) = \{h : s_i(h) = s_i\}$, and it is

assumed that they all share the same set of legal actions $\mathcal{A}_i(s_i(h)) = \mathcal{A}_i(h)$. For simplicity we often drop the subscript $i$ for an information set $s$ when the player is implied.

A player's *strategy* $\pi_i$ is a function mapping from an information set to a probability distribution over actions. A *strategy profile* $\pi$ is a tuple $(\pi_1, \ldots, \pi_N)$. All players other than $i$ are denoted $-i$, and their strategies are jointly denoted $\pi_{-i}$. A strategy for a history $h$ is denoted $\pi_i(h) = \pi_i(s_i(h))$ and $\pi(h)$ is the corresponding strategy profile. When a strategy $\pi_i$ is learned through reinforcement learning (RL), we refer to the learned strategy as a *policy*.

The *expected value (EV)* $v_i^\pi(h)$ for player $i$ is the expected sum of future utilities for player $i$ in history $h$, when all players play strategy profile $\pi$. The EV for an information set $s_i$ is denoted $v_i^\pi(s_i)$ and the EV for the entire game is denoted $v_i(\pi)$. A *two-player zero-sum* game has $v_1(\pi) + v_2(\pi) = 0$ for all strategy profiles $\pi$. The EV for an action in an information set is denoted $v_i^\pi(s_i, a_i)$. A *Nash equilibrium (NE)* is a strategy profile such that, if all players played their NE strategy, no player could achieve higher EV by deviating from it. Formally, $\pi^*$ is a NE if $v_i(\pi^*) = \max_{\pi_i} v_i(\pi_i, \pi_{-i}^*)$ for each player $i$.

The *exploitability* $e(\pi)$ of a strategy profile $\pi$ is defined as $e(\pi) = \sum_{i \in \mathcal{N}} \max_{\pi_i'} v_i(\pi_i', \pi_{-i})$. A *best response (BR)* strategy $\mathbb{BR}_i(\pi_{-i})$ for player $i$ to a strategy $\pi_{-i}$ is a strategy that maximally exploits $\pi_{-i}$: $\mathbb{BR}_i(\pi_{-i}) \in \arg\max_{\pi_i} v_i(\pi_i, \pi_{-i})$. An *$\epsilon$-best response ($\epsilon$-BR)* strategy $\mathbb{BR}_i^\epsilon(\pi_{-i})$ for player $i$ to a strategy $\pi_{-i}$ is a strategy that is at most $\epsilon$ worse for player $i$ than the best response: $v_i(\mathbb{BR}_i^\epsilon(\pi_{-i}), \pi_{-i}) \geq v_i(\mathbb{BR}_i(\pi_{-i}), \pi_{-i}) - \epsilon$. An *$\epsilon$-Nash equilibrium ($\epsilon$-NE)* is a strategy profile $\pi$ in which, for each player $i$, $\pi_i$ is an $\epsilon$-BR to $\pi_{-i}$.

## 2.1 Counterfactual Regret Minimization (CFR)

In this section we review the *counterfactual regret minimization (CFR)* framework. All superhuman poker AIs have used advanced variants of the framework as part of their architectures (Bowling et al., 2015; Brown & Sandholm, 2018; 2019). CFR is also the basis of several reinforcement learning algorithms described in Section B. We will leverage and extend the CFR framework in the rest of the paper. We will start by reviewing the framework.

Define $\eta^\pi(h) = \prod_{a \in h} \pi(a)$ to be the reach weight of joint policy $\pi$ to reach history $h$, and $z$ is a terminal history. Define $\eta^\pi(h, z) = \frac{\eta^\pi(z)}{\eta^\pi(h)}$ to be the reach weight of joint policy $\pi$ to reach terminal history $z$ from history $h$. Define $Z$ to be the set of all terminal histories. Define $Z(s) \subseteq Z$ to be the set of terminal histories $z$ that can be reached from information state $s$ and define $z[s]$ to be the unique history $h \in s$ that is a subset of $z$. Define

$$v_i(\pi, h) = \sum_{z \sqsupseteq h} \eta^\pi(h, z) u_i(z) \tag{1}$$

to be the expected value under $\pi$ for player $i$ having reached $h$. Note that this value function takes as input the full-information history $h$ and not an information set. Define

$$v_i^c(\pi, s) = \sum_{z \in Z(s)} \eta_{-i}^\pi(z[s]) \eta^\pi(z[s], z) u_i(z) = \sum_{h \in s} \eta_{-i}^\pi(h) v_i(\pi, h) \tag{2}$$

to be the *counterfactual value* for player $i$ at state $s$ under the joint strategy $\pi$. Define the strategy $\pi_{s \to a}$ to be a modified version of $\pi$ where $a$ is played at information set $s$, and the counterfactual state-action value $q_i^c(\pi, s, a) = v_i^c(\pi_{s \to a}, s)$. Similarly, define the history-action value $q_i(\pi, h, a) = v_i(\pi_{h \to a}, h)$. For any state $s$, strategy $\pi$, and action $a \in \mathcal{A}(s)$, one can define a local *counterfactual regret* for not switching to playing $a$ at $s$ as $r^c(\pi, s, a) = q_i^c(\pi, s, a) - v_i^c(\pi, s)$. Counterfactual regret minimization (CFR) (Zinkevich et al., 2008a) is a strategy iteration algorithm that produces a sequence of policies: $\{\pi_1, \pi_2, \cdots, \pi_T\}$. Each policy $\pi_{t+1}(s)$ is derived directly from a collection of cumulative regrets $R_i^T(s, a) = \sum_{t=1}^T r^c(\pi_t, s, a)$, for all $a \in \mathcal{A}(s)$ using regret-matching (Hart & Mas-Colell, 2000). *Total regret* for player $i$ in the entire game is defined as $R_i^T = \max_{\pi_i'} \sum_{t=1}^T v_i(\pi_i', \pi_{-i}^t) - v_i(\pi_i^t, \pi_{-i}^t)$. In two-player zero-sum games, the average policy $\bar{\pi}_T(s) = \frac{\sum_{t=1}^T \eta_i^{\pi_t}(s) \pi_t(s)}{\sum_{t=1}^T \eta_i^{\pi_t}(s)}$ converges to an approximate Nash equilibrium at a rate of $e(\bar{\pi}_T) \leq O(1/\sqrt{T})$.

## 2.2 Monte Carlo Counterfactual Regret Minimization (MCCFR)

In the standard CFR algorithm, the quantities required to produce new policies in Equations 1 and 2 require full traversals of the game to compute exactly. Monte Carlo CFR (Lanctot et al., 2009) is a *stochastic* version of CFR which instead *estimates* these quantities. In particular, MCCFR uses a sampling approach which specifies a distribution over blocks $Z_j$ of terminal histories such that $\cup_j Z_j = \mathcal{Z}$, the set of terminal histories. Upon sampling a block $j$, a certain *sampled counterfactual value* $\hat{v}^c(\pi, s \mid j)$ (defined in detail later in this section) is computed for all prefix histories that occur in $Z_j$. Then, estimated regrets are accumulated and new policies derived as in CFR. The main result is that $\mathbb{E}[\hat{v}^c(\pi, s \mid j)] = v^c(\pi, s)$, so MCCFR is an unbiased approximation of CFR, and inherits its convergence properties albeit under a probabilistic guarantee.

Blocks are sampled via sampling policy $\tilde{\pi}$ which is commonly a function of the players' joint policy $\pi$. Two sampling variants were defined in the original MCCFR paper: *outcome sampling* (OS-MCCFR) and *external sampling* (ES-MCCFR). External sampling samples only the opponent (and chance's) choices; hence, it requires a forward model of the game to recursively traverse over all of the subtrees under the player's actions. Outcome sampling is the most extreme sampling variant where blocks consist of a single terminal history: it is the only model-free variant of MCCFR compliant with the standard reinforcement learning loop where the agent learns entirely from experience with the environment. The OS-MCCFR counterfactual value estimator when the opponent samples from their current policy as is commonly done is given as follows:

$$\hat{v}_i(\pi, s|z) = \frac{\eta^{\pi^{-i}}(z[s])\eta^{\pi}(z[s], z)u_i(z)}{\eta^{\tilde{\pi}}(z)} = \frac{1}{\eta^{\tilde{\pi}_i}(z[s])} \frac{\eta^{\pi_i}(z[s], z)}{\eta^{\tilde{\pi}_i}(z[s], z)} u_i(z) \tag{3}$$

The importance sampling term that is used to satisfy the unbiasedness of the values can have a significant detrimental effect on the convergence rate Gibson et al. (2012). Variance reduction techniques provide some empirical benefit Schmid et al. (2019); Davis et al. (2019), but have not been evaluated on games with long trajectories where the importance corrections have their largest impact.

## 2.3 Deep Counterfactual Regret Minimization

Deep CFR (Brown et al., 2019; Steinberger, 2019; Li et al., 2019) is a method that uses neural networks to scale MCCFR to large games. Deep CFR performs external sampling MCCFR and trains a regret network $R_i(s, a|\psi)$ on a replay buffer of information sets and estimated cumulative regrets. The regret network is trained to approximate the cumulative regrets seen so far at that information state. The estimated counterfactual regrets are computed the same as in MCCFR, namely using outcome sampling or external sampling.

## 2.4 DREAM

DREAM (Steinberger et al., 2020) builds on Deep CFR and approximates OS-MCCFR with deep neural networks. Like Deep CFR, it trains a regret network $R_i(s, a|\psi)$ on a replay buffer of information sets and estimated cumulative regrets. Additionally, in order to limit the high variance of OS-MCCFR, DREAM uses a learned history value function $q_i(\pi, h, a|\theta)$ and uses it as a baseline (Schmid et al., 2019). While the baseline helps remove variance in the estimation of future utility, in order to remain unbiased DREAM must use importance sampling as in OS-MCCFR. We show empirically that variance of the DREAM estimator of the counterfactual value, although lower than OS-MCCFR, will often be quite high, even in small games and with an oracle history value function. This high variance estimator might make neural network training very difficult. In contrast, ESCHER has no importance sampling term and instead directly uses the learned history value function $q_i(\pi, h, a|\theta)$ to estimate regrets.

## 3 ESCHER

In this section we define our proposed algorithm, *ESCHER*, where we use a learned history value function $v_i(\pi, h|\theta)$ to estimate regret. This history value function is a neural network trained on on-policy rollouts to predict future reward. In our theoretical analysis we model the learned history

value function to be equal to the exact history value $v_i(\pi, h)$ plus an additional error. In our theoretical analysis we show that ESCHER is sound and converges to a Nash equilibrium with high probability.

As shown in Equation 3, the OS-MCCFR estimator can be seen as containing two separate terms. The first $1/\eta^{\tilde{\pi}_i}(z[s])$ term ensures that each information set is updated equally often in expectation. The second $\eta^{\pi_i}(z[s], z)u_i(z)/\eta^{\tilde{\pi}_i}(z[s], z)$ term is an unbiased estimator of the history value $v_i(\pi, z[s])$. In DREAM, the second term gets updated by a bootstrapped baseline to reduce variance. But since the baseline is not perfect in practice, as we show in our ablations, this term still induces high variance, which prevents the regret network from learning effectively. The main idea behind ESCHER is to remove the first reach weighting term by ensuring that the sampling distribution for the update player remains fixed across iterations, and to replace the second term with a history value function $v_i(\pi, z[s]|\theta)$.

Our method is built on Deep CFR. In particular, like Deep CFR, we traverse the game tree and add this experience into replay buffers. The first replay buffer stores information states and instantaneous regret estimates is used to train a regret network $R_\psi(s, a)$ that is trained to estimate the cumulative regret at a given information set. This replay buffer is filled with information sets from trajectories gathered by following the sampling distribution in Equation 5. The regret network $R_i(s, a|\psi)$ is reinitialized and trained from scratch on the entire replay buffer to estimate the average estimated counterfactual regret at every information set via regression. Since regret matching is scale invariant, the policy computed from average counterfactual regret is equivalent to the policy computed from total counterfactual regret. Similar to Deep CFR, each player's current policy $\pi_i$ is given by performing regret matching (described in Equation 7) on the output of the current regret network $R_i(s, a|\psi)$.

The second replay buffer stores histories and terminal utilities and is used to train the value network $v_\theta$ to estimate the expected utility for both players when both players are at that history and play from their current policies. The value network $v_\theta$ is reinitialized after every iteration, and is trained by predicting future reward from on-policy trajectories. In particular, to avoid the issue that the regret-matching policy does not generally put strictly positive mass on all actions, the on-policy sampling is performed by sampling from a weighted mixture of $0.99 \times$ the current policy and $0.01 \times$ the uniformly-random policy. Lastly, the third replay buffer stores information states and actions taken by the policy $\pi$ and uses that data to train an average policy network $\bar{\pi}_\phi$ that approximates the average policy across all iterations. It is this average policy that has no regret and converges to an approximate Nash equilibrium in self play.

Unlike Deep CFR and DREAM, which use the terminal utility and sampling probabilities from the current trajectory to estimate the value, in ESCHER the instantaneous regret estimates are estimated using the current history value function $v_i(\pi, h|\theta)$ alone. Since we only update regret on information states visited during the trajectory, our estimator is zero on all other information sets. Formally, we define our estimator for the counterfactual regret as follows:

$$\hat{r}_i(\pi, s, a|z) = \begin{cases} v_i(\pi, z[s]a|\theta) - v_i(\pi, z[s]|\theta) & \text{if } z \in Z(s) \\ 0 & \text{otherwise} \end{cases} \tag{4}$$

ESCHER samples from the opponent's current strategy when it is their turn but samples from a fixed strategy that roughly visits every information set equally likely when it is the update player's turn. As a result, the expected value of the history value is equal to the counterfactual value scaled by a term that weights certain information sets up or down based on the fixed sampling policy. Formally, define the fixed sampling policy $b_i(s, a)$ to be any policy that remains constant across iterations and puts positive probability on every action. This fixed sampling policy can be one of many distributions such as one that samples uniformly over available actions at every information set. In games with an equal number of actions at every information set, the uniform policy will sample all information sets at each level of the tree equally likely. An interesting open research direction is finding good fixed sampling policy. In this paper, our fixed sampling policy uniformly samples over actions, which is somewhat similar to the robust sampling technique introduced in Li et al. (2019). When updating player $i$, we construct a joint fixed sampling policy $\tilde{\pi}^i(s, a)$ to be

$$\tilde{\pi}^i(s, a) = \begin{cases} b_i(s, a) & \text{if it's the update player } i\text{'s turn} \\ \pi_{-i}(s, a) & \text{otherwise} \end{cases} \tag{5}$$

---

**Algorithm 1:** ESCHER

---

1    Initialize history value function $q$
2    Initialize policy $\pi_i$ for both players
3    **for** $t = 1, ..., T$ **do**
4       Refill value replay buffer with on-policy data from $\pi$
5       Retrain history value function on data from value replay buffer
6       Reinitialize regret networks $R_0, R_1$
7       **for** *update player* $i \in \{0, 1\}$ **do**
8         **for** $P$ *trajectories* **do**
9           Get trajectory $\tau$ using sampling distribution (Equation 5)
10           **for** *each state* $s \in \tau$ **do**
11             **for** *each action* $a$ **do**
12               Estimate immediate cf-regret $\hat{r}(\pi, s, a|z) = v_i(\pi, z[s]a|\theta) - v_i(\pi, z[s]|\theta)$
13             Add $(s, \hat{r}(\pi, s))$ to cumulative regret buffer
14             Add $(s, a')$ to average policy buffer where $a'$ is action taken at state $s$
15         Train regret network $R_i$ on cumulative regret buffer
16    Train average policy network $\bar{\pi}_\phi$ on average policy buffer
17    **return** average policy network $\bar{\pi}_\phi$

---

We use a fixed sampling policy because it allows us to remove any importance sampling in our estimator. Unlike Deep CFR and DREAM which must divide by the current player's reach probability to remain unbiased, our method does not use importance sampling but total average regret is still guaranteed to converge to zero. We describe our algorithm in Algorithm 1. Highlighted in blue are the differences between our algorithm and Deep CFR. Namely, we train a history value function and use it to estimate counterfactual regret.

## 3.1 THEORETICAL RESULTS

In the following, assume that our learned value function $v_i(\pi, h|\theta)$ is equal to the exact value function $v_i(\pi, h)$. In the appendix we analyze the case where the learned value function is inaccurate. If we sample from $\tilde{\pi}^i$ when updating player $i$, then the expected value of our counterfactual regret estimator is:

$$
\begin{aligned}
\mathbf{E}_{z \sim \tilde{\pi}^i}[\hat{r}_i(\pi, s, a|z)] &= \sum_{z \in Z} \eta^{\tilde{\pi}^i}(z)[\hat{r}_i(\pi, s, a|z)] \\
&= \sum_{z \in Z(s)} \eta^{\tilde{\pi}^i}(z)[v_i(\pi, z[s]a) - v_i(\pi, z[s])] \\
&= \sum_{h \in s} \sum_{z \sqsupseteq h} \eta^{\tilde{\pi}^i}(z)[v_i(\pi, z[s]a) - v_i(\pi, z[s])] \\
&= \sum_{h \in s} \eta^{\tilde{\pi}^i}(h)[v_i(\pi, ha) - v_i(\pi, h)] \\
&= \eta_i^{\tilde{\pi}^i}(s) \sum_{h \in s} \eta_{-i}^{\pi}(h)[v_i(\pi, ha) - v_i(\pi, h)] \\
&= w(s)[v_i^c(\pi, s, a) - v_i^c(\pi, s)] = w(s)r^c(\pi, s, a)
\end{aligned}
\tag{6}
$$

Where for $h, h' \in s$, $\eta_i^b(h) = \eta_i^b(h') = \eta_i^b(s) =: w(s)$ is the reach probability for reaching that infostate for player $i$ via the fixed sampling distribution. Unlike the estimator in Deep CFR and DREAM, our estimator has no importance sampling terms, and as a result has much lower variance. When all information sets are visited by the sampling distribution with equal probability, then ESCHER is perfectly unbiased. The correctness of our method is established by the next theorem, whose proof can be found in Appendix A. As shown in the proof, the regret of our method is bounded with high probability by a term that is inversely proportional to the minimum over information sets $s$ of $w(s)$. Therefore, our theory suggests that the balanced sampling distribution is the optimal sampling distribution, but in practice other sampling distributions might perform better. In our experiments we approximate the balanced distribution with uniform sampling over actions.

| Game | ESCHER (Ours) | Ablation 1 | Ablation 2 | DREAM |
|---|---|---|---|---|
| Phantom Tic-Tac-Toe | $(2.6 \pm 0.1) \times 10^{-1}$ | $(4.1 \pm 0.7) \times 10^{1}$ | $(1.4 \pm 0.4) \times 10^{7}$ | $(4.6 \pm 1.0) \times 10^{7}$ |
| Dark Hex 4 | $(1.8 \pm 0.1) \times 10^{-1}$ | $(1.3 \pm 0.9) \times 10^{2}$ | $(3.1 \pm 1.7) \times 10^{8}$ | $(2.8 \pm 2.0) \times 10^{8}$ |
| Dark Hex 5 | $(1.3 \pm 0.1) \times 10^{-1}$ | $(3.3 \pm 1.6) \times 10^{2}$ | $(2.0 \pm 0.6) \times 10^{5}$ | $(5.3 \pm 3.9) \times 10^{8}$ |

| Game | ESCHER (Ours) | Ablation 2 | DREAM | OS-MCCFR |
|---|---|---|---|---|
| Leduc | $(5.3 \pm 0.0) \times 10^{0}$ | $(3.3 \pm 0.7) \times 10^{2}$ | $(2.8 \pm 0.0) \times 10^{2}$ | $(2.2 \pm 0.0) \times 10^{3}$ |
| Battleship | $(1.4 \pm 0.0) \times 10^{0}$ | $(7.1 \pm 0.3) \times 10^{2}$ | $(1.2 \pm 0.0) \times 10^{3}$ | $(2.4 \pm 0.0) \times 10^{3}$ |
| Liar's Dice | $(9.0 \pm 0.1) \times 10^{-1}$ | $(7.8 \pm 0.9) \times 10^{1}$ | $(4.0 \pm 0.8) \times 10^{2}$ | $(1.2 \pm 0.1) \times 10^{3}$ |

Table 1: These results track the average variance of the regret estimator of each algorithm over all iterations. The top table shows results of deep algorithms on large games. Ablation 1 is ESCHER but with a bootstrapped baseline, and ablation 2 is ESCHER but with reach weighting. The bottom table shows tabular versions of the algorithms with oracle value functions on small games. Because ESCHER does not use importance sampling, the variance of its estimator is orders of magnitude smaller than baselines. Colors scale with the ratio of the minimum value in each row, according to the logarithmic color scale ☐ .
1   10   100  ≥ 1000

| Algorithm | History value function | Boostrapped baseline | Importance sampling |
|---|---|---|---|
| ESCHER (Ours) | ✓ | ✗ | ✗ |
| Ablation 1 | ✓ | ✓ | ✗ |
| Ablation 2 | ✓ | ✗ | ✓ |
| DREAM / VR-MCCFR | ✓ | ✓ | ✓ |
| Deep CFR / OS-MCCFR | ✗ | ✗ | ✓ |

Table 2: ESCHER is different from DREAM in two important ways. First, ESCHER does not use importance sampling (Lanctot et al., 2009; Brown et al., 2019; Steinberger et al., 2020). Second, ESCHER does not estimate counterfactual values for sampled actions via a bootstrapped baseline (Schmid et al., 2019; Steinberger et al., 2020). Our ablations show that both improvements are necessary.

**Theorem 1.** *Assume a fixed sampling policy that puts positive probability on every action. For any $p \in (0, 1)$, with probability at least $1 - p$, the regret accumulated by each agent learning using the tabular algorithm ESCHER (Algorithm 2) is upper bounded by $O(\sqrt{T} \cdot \text{poly} \log(1/p))$, where the $O(\cdot)$ notation hides game-dependent and sampling-policy-dependent constants.*

In the appendix we extend the analysis to the case of approximate history value function, and give a bound with an explicit dependence on the magnitude of the approximation error.

## 4 RESULTS

We compare the variance of the counterfactual value estimates from ESCHER, DREAM, and ablations in Table 1. Variance is computed over the set of all counterfactual regret values estimated in a single iteration. The results in the table are the average of each iteration's variance over the first five iterations of training. The top table shows results of deep algorithms on large games. A summary of the different ablations is given in Table 2. From this experiment we can see that because ESCHER does not use importance sampling, the variance of its estimator is orders of magnitude smaller than baselines. Also, we see that even ablation 1, which does not use importance sampling, has high variance. This is because when the history value function is not exact, the bootstrapping method recursively divides by the sampling probability on sampled actions. We see that this higher variance

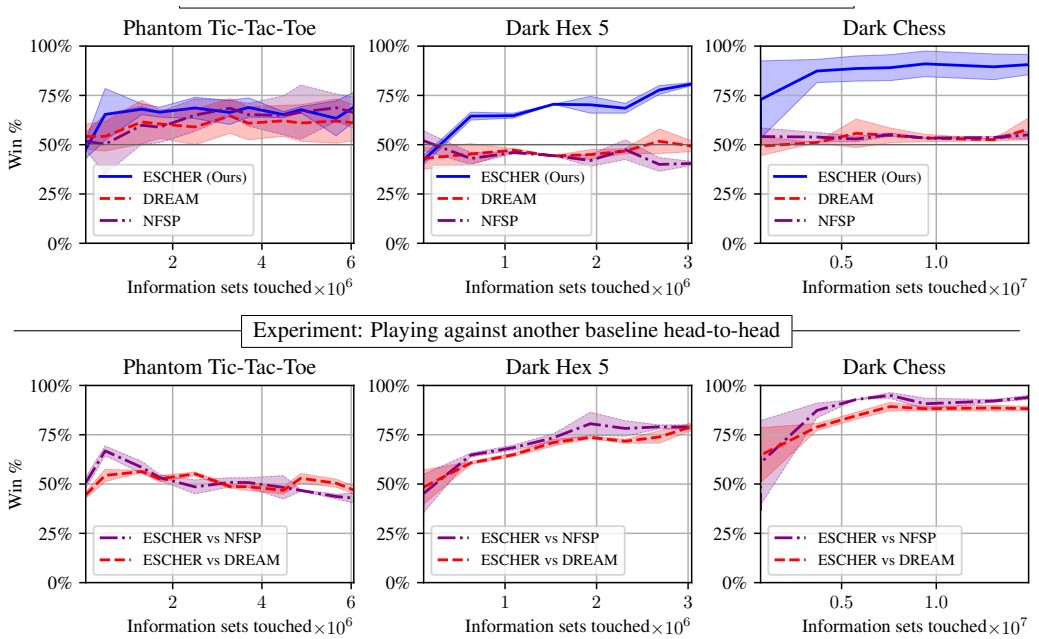

Figure 1: ESCHER is competitive with NFSP and DREAM in Phantom Tic-Tac-Toe. But as the size of the game increases, ESCHER performs increasingly better than both DREAM and NFSP. In Dark Chess, ESCHER beats DREAM and NFSP in over $90\%$ of the matches.

indeed leads to worse performance for DREAM and these ablations in Figure 2. Therefore, both improvements of ESCHER over DREAM are necessary.

We compare our method to DREAM and NFSP, the most popular baselines that are also open-source, on the games of Phantom Tic Tac Toe (TTT), Dark Hex, and Dark Chess. Dark chess is a popular game among humans under the name *Fog of War Chess* on the website chess.com, and has emerged as a benchmark task (Zhang & Sandholm, 2021). All of these games are similar in that they are both imperfect information versions of perfect-information games played on square boards. Phantom TTT is played on a $3 \times 3$ board while dark hex 4 is played on a size $4 \times 4$ board and dark hex 5 is played on a size $5 \times 5$ board. Because these games are large, we are not able to compare exact exploitability so instead we compare performance through head-to-head evaluation. Results are shown in Figure 1, where the x axis tracks the number of information sets visited during training. We see that our method is competitive with DREAM and NFSP on Phantom TTT. On the larger game of Dark Hex 5, ESCHER beats DREAM and NFSP head to head and also scores higher against a random opponent. Moving to the largest game of Dark Chess, we see that ESCHER beats DREAM and NFSP head to head over $90\%$ of the time and also is able to beat a random opponent while DREAM and NFSP are no better than random.

## 5 DISCUSSION: LIMITATIONS AND FUTURE RESEARCH

Our method has a number of ways it can be improved. First, it requires two separate updates in one iteration. Perhaps the method could be more efficient if only on-policy data were used. Second, our method, like Deep CFR, DREAM, and NFSP, trains neural networks on large replay buffers of past experience. Unlike RL algorithms like DQN (Mnih et al., 2015), these replay buffers must record all data ever seen in order to learn an average. This can be a problem when the amount of data required is much larger than the replay buffer memory. Third, we do not use various implementation details that help performance in Deep CFR such as weighting by the sum of the reach probabilities over all iterations. Finally, our method uses separate data to train the value function. Our method could be made much more efficient by also using the data generated for training the policy to also train the value function.

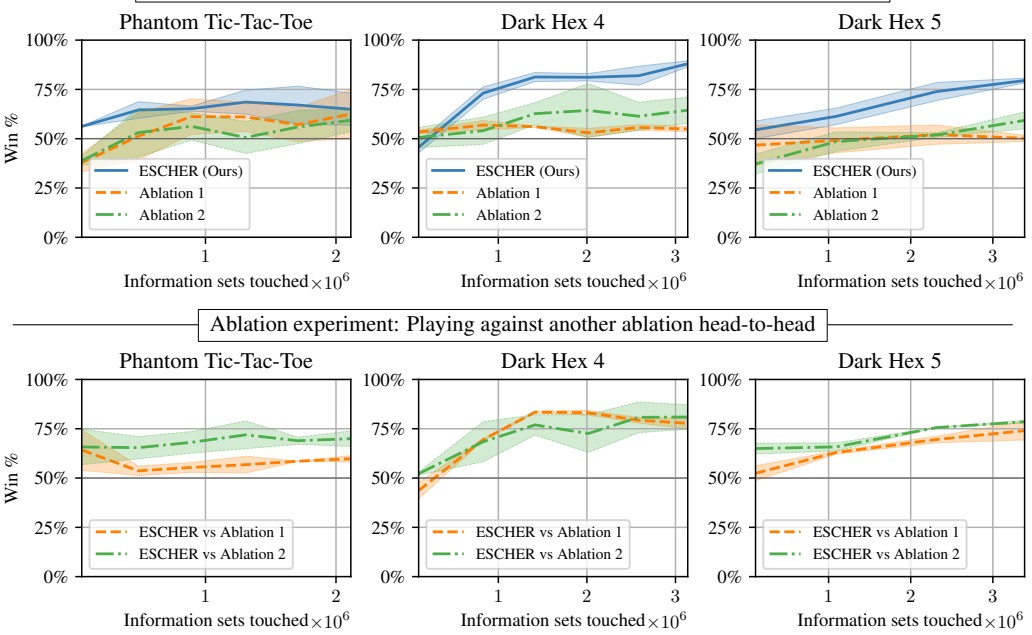

Figure 2: Ablation study on ESCHER. As summarized in Table 2, "Ablation 1" is ESCHER but with a bootstrapped history-value baseline, while "Ablation 2" is ESCHER but with reach weighting. Since ESCHER with a bootstrapped history-value baseline and reach weighting is equivalent to DREAM, these results show that both changes to DREAM are necessary for ESCHER to work in large games.

One direction of future research is finding optimal sampling distributions. In our method we use the uniform distribution over actions as our fixed sampling distribution, but this can be far from optimal. In principle any distribution that remains fixed will guarantee the method to converge with high probability. One possible direction would be to try to estimate the theoretically optimal balanced distribution. Other, less principled, methods such as using the average policy might work well in practice as well (Burch et al., 2012). Another direction is in connecting this work with the reinforcement learning literature. Similar to reinforcement learning, we learn a Q value and a policy, and there are many techniques from reinforcement learning that are promising to try in this setting. For example, although we learned the value function simply through Monte-Carlo rollouts, one could use bootstrapping-based methods such as TD-$\lambda$ (Sutton, 1988) and expected SARSA (Rummery & Niranjan, 1994). The policy might be able to be learned via some sort of policy gradient, similar to QPG (Srinivasan et al., 2018), NeuRD (Hennes et al., 2020), and F-FoReL (Perolat et al., 2021).

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

## A  PROOFS

### A.1  KNOWN RESULTS ABOUT REGRET MATCHING (RM)

In this section, we recall the definition and the regret bound of the Regret Matching (RM) regret minimization algorithm for probability simplices. In the interest of keeping the paper as self-contained as possible, we propose a proof for the result. For a deeper treatment of RM, we invite the reader to consult the works of Hart & Mas-Colell (2000); Gordon (2007); Farina et al. (2021).

We start by recalling the definition of RM. Let $A$ be a set of discrete actions and $\Delta(A)$ the simplex of probability distributions over $A$. Regret Matching operates by keeping a tally $R_a^t$ of the regret accumulated up to each time $t$ compared to always selecting each action $a$, and picks the next distribution according to

$$\boldsymbol{x}^t(a) := \begin{cases} \dfrac{\max\{0, R_a^t\}}{\sum_{a' \in A} \max\{0, R_{a'}^t\}} & \text{if } \sum_{a' \in A} \max\{0, R_{a'}^t\} > 0 \\ \dfrac{1}{|A|} & \text{otherwise.} \end{cases} \tag{7}$$

Upon observing the utility vector $\boldsymbol{g}^t \in \mathbb{R}^A$, the tally of the regret for each action is updated correspondingly as

$$R_a^{t+1} := R_a^t + \boldsymbol{g}^t(a) - \sum_{a' \in A} \boldsymbol{g}^t(a') \boldsymbol{x}^t(a'). \tag{8}$$

(at time $t = 0$, $R_a^0 = 0$ for all $a \in A$).

**Proposition 1.** *Let $M > 0$, and let $\boldsymbol{g}^t$ be an arbitrary (possibly adversarially picked) sequence of utility vectors with $|\boldsymbol{g}^t(a)| \le M$ for all $a \in A$ received by RM. The regret*

$$R^T := \max_{a \in A} R_a^T = \max_{a \in A} \left\{ \sum_{t=1}^T \boldsymbol{g}^t(a) - \sum_{t=1}^T \sum_{a' \in A} \boldsymbol{g}^t(a') \boldsymbol{x}^t(a') \right\}$$

*accumulated up to any time $T$ by RM satisfies*

$$R^T \le M \sqrt{|A|T}.$$

*Proof.* The proof hinges on the following observation. Fix any time $t$.

- If $\sum_{a' \in A} \max\{0, R_{a'}^t\} \le 0$, then $R_{a'}^t \le 0$ for all $a' \in A$, and so trivially

$$\sum_{a \in A} \max\{0, R_a^t\} \left( \boldsymbol{g}^t(a) - \sum_{a' \in A} \boldsymbol{g}^t(a') \boldsymbol{x}^t(a') \right) = 0.$$

- On the other hand, if $\sum_{a' \in A} \max\{0, R_{a'}^t\} > 0$ then

$$\boldsymbol{x}^t(a) = \frac{\max\{0, R_a^t\}}{\sum_{a' \in A} \max\{0, R_{a'}^t\}}$$

and so

$$\sum_{a \in A} \max\{0, R_a^t\} \left( \boldsymbol{g}^t(a) - \sum_{a' \in A} \boldsymbol{g}^t(a') \boldsymbol{x}^t(a') \right)$$

$$= \left( \sum_{a \in A} \max\{0, R_a^t\} \boldsymbol{g}^t(a) \right) - \left( \sum_{a \in A} \max\{0, R_a^t\} \right) \sum_{a' \in A} \boldsymbol{g}^t(a') \boldsymbol{x}^t(a')$$

$$= 0.$$

So, in either case, at all times $t$,

$$\sum_{a \in A} \max\{0, R_a^t\} \left( \boldsymbol{g}^t(a) - \sum_{a' \in A} \boldsymbol{g}^t(a') \boldsymbol{x}^t(a') \right) = 0. \tag{9}$$

Now, observe that for all $t \geq 2$

$$
\sum_{a \in A} \max\{0, R_a^{t+1}\}^2 \leq \sum_{a \in A} (R_a^{t+1} - \min\{0, R_a^t\})^2
$$

$$
= \sum_{a \in A} \left( R_a^t - \min\{0, R_a^t\} + \boldsymbol{g}^t(a) - \sum_{a' \in A} \boldsymbol{g}^t(a')\boldsymbol{x}^t(a') \right)^2 \qquad \text{(from (8))}
$$

$$
= \sum_{a \in A} \left( \max\{0, R_a^t\} + \boldsymbol{g}^t(a) - \sum_{a' \in A} \boldsymbol{g}^t(a')\boldsymbol{x}^t(a') \right)^2
$$

$$
= \sum_{a \in A} \max\{0, R_a^t\}^2 + \left( \boldsymbol{g}^t(a) - \sum_{a' \in A} \boldsymbol{g}^t(a')\boldsymbol{x}^t(a') \right)^2
$$

$$
+ 2 \sum_{a \in A} \max\{0, R_a^t\} \left( \boldsymbol{g}^t(a) - \sum_{a' \in A} \boldsymbol{g}^t(a')\boldsymbol{x}^t(a') \right)
$$

$$
= \sum_{a \in A} \max\{0, R_a^t\}^2 + \left( \boldsymbol{g}^t(a) - \sum_{a' \in A} \boldsymbol{g}^t(a')\boldsymbol{x}^t(a') \right)^2 \qquad \text{(from (9))}
$$

$$
\leq \sum_{a \in A} \max\{0, R_a^t\}^2 + |A| M^2,
$$

where the last inequality follows from the fact that by assumption each utility has absolute value at most $M$, that is, $|\boldsymbol{g}^t(a)| \leq M$. Using the fact that $R_a^0 = 0$ for all $a \in A$, the previous recursive inequality leads to

$$
\sum_{a \in A} \max\{0, R_a^T\}^2 \leq T|A|M^2 \qquad \forall T \in \{1, 2, \dots\}.
$$

From the previous inequality we finally conclude that

$$
R^T = \max_{a \in A} \max\{0, R_a^T\} \leq \sqrt{\sum_{a \in A} \max\{0, R_a^T\}^2} \leq M\sqrt{|A|T},
$$

as we wanted to show. $\qquad\square$

## A.2 ANALYSIS OF ESCHER

We start by recalling a central theorem connecting regret to counterfactual regret (see, *e.g.*, Zinkevich et al. (2008b); Farina et al. (2019b;a)).

**Proposition 2.** *Fix any player $i$, and let*

$$
R_s^T := \max_{\hat{a} \in A_s} \sum_{t=1}^{T} r_i^c(\pi^t, s, \hat{a}) = \max_{\hat{a} \in A_s} \sum_{t=1}^{T} q_i^c(\pi^t, s, \hat{a}) - v_i^c(\pi^t, s)
$$

*be the counterfactual regret accumulated up to time $T$ by the regret minimizer local at each information set $s$. Then, the regret*

$$
R_i^T := \max_{\hat{\pi}_i} \sum_{t=1}^{T} v_i(\hat{\pi}_i, \pi_{-i}^t) - v_i(\pi_i^t, \pi_{-i}^t)
$$

*accumulated by the policies $\pi^t$ on the overall game tree satisfies*

$$
R_i^T \leq \sum_{s} \max\{R_s^T, 0\}.
$$

We can now use a modification of the argument by Farina et al. (2020) to bound the degradation of regret due to the use of an estimator of the counterfactual regrets. However, our analysis requires some modifications compared to that of Farina et al. (2020), in that ESCHER introduces estimation at the level of counterfactuals, while the latter paper introduces estimation at the level of the game utilities.

---

**Algorithm 2:** Tabular ESCHER with Oracle Value Function

---

1 **for** $t = 1, ..., T$ **do**
2    **for** *update player* $i \in \{0, 1\}$ **do**
3       Sample trajectory $\tau$ using sampling distribution $\tilde{\pi}^i$ (Equation 5)
4       **for** *each state* $s \in \tau$ **do**
5          **for** *each action* $a$ **do**
6             Estimate immediate regret vector $\hat{r}(\pi, s, a|z) = q_i(\pi, z[s], a) - v_i(\pi, z[s])$
7             Update total estimated regret of action $a$ at infostate $s$:
              $\hat{R}(s, a) = \hat{R}(s, a) + \hat{r}(\pi, s, a|z)$
8             Update $\pi_i(s, a)$ via regret matching (Equation 7) on total estimated regret
9 **return** average policy $\bar{\pi}$

---

**Theorem 1.** *Assume a fixed sampling policy that puts positive probability on every action. For any $p \in (0, 1)$, with probability at least $1 - p$, the regret accumulated by each agent learning using the tabular algorithm ESCHER (Algorithm 2) is upper bounded by $O(\sqrt{T} \cdot \text{poly} \log(1/p))$, where the $O(\cdot)$ notation hides game-dependent and sampling-policy-dependent constants.*

*Proof.* As shown in Section 3, for any information set $s$ the counterfactual regret estimators $\hat{r}_i(\pi, s, a|h)$ are unbiased up to a time-independent multiplicative factor; specifically,

$$\mathbf{E}_{h \sim \tilde{\pi}^i}[\hat{r}_i(\pi, s, a|h)] = w(s)r_i(\pi, s, a)$$

for all actions $a$ available to player $i$ at world states in $s$. Hence, for each $a \in A_s$ we can construct the martingale difference sequences

$$X_a^t := w(s)r_i(\pi^t, s, a) - \hat{r}_i(\pi^t, s, a|h).$$

Clearly, $X_t^a$ is bounded, with $|X_t^a|$ upper bounded by (twice) the range $D$ of payoffs of player $i$. Hence, from the Azuma-Hoeffding inequality, we obtain that the regret $R_s^T$ accumulated by the local policies produced by ESCHER with respect to the correct counterfactuals satisfies, for all $p \in (0, 1)$

$$\mathbf{P}\left[\sum_{t=1}^{T} X_a^t \leq 2D\sqrt{2T \log \frac{1}{p}}\right] \geq 1 - p,$$

Using a union bound on the actions, we can therefore write

$$\mathbf{P}\left[\max_a \sum_{t=1}^{T} X_a^t \leq 2D\sqrt{2T \log \frac{|A_s|}{p}}\right] \geq 1 - p,$$

The left-hand side in the probability can be expanded as follows:

$$\max_a \sum_{t=1}^{T} X_a^t = \max_a \left\{ w(s)\sum_{t=1}^{T} r_i(\pi^t, s, a) - \sum_{t=1}^{T} \hat{r}_i(\pi^t, s, a|h) \right\}$$

$$\geq \max_a \left\{ w(s)\sum_{t=1}^{T} r_i(\pi^t, s, a) \right\} - \max_a \left\{ \sum_{t=1}^{T} \hat{r}_i(\pi^t, s, a|h) \right\}$$

$$\geq w(s)R_s^T - D\sqrt{|A_s|T},$$

where the last inequality follows from the fact that the regret cumulated by regret matching (which is run on the regret estimates $\hat{r}_i$) is upper bounded by $D\sqrt{|A_s|T}$ (see Proposition 1). Hence, we can write

$$\mathbf{P}\left[w(s)R_s^T - D\sqrt{|A_s|T} \leq 2D\sqrt{2T \log \frac{|A_s|}{p}}\right] \geq 1 - p$$

$$\iff \quad \mathbf{P}\left[R_s^T \leq \frac{D}{w(s)}\sqrt{|A_s|T} + \frac{2D}{w(s)}\sqrt{2T \log \frac{|A_s|}{p}}\right] \geq 1 - p,$$

where in the second step we used the hypothesis that $w(s) > 0$ for all $s$. Since the right-hand size inside of the probability is non-negative, we can further write

$$\mathbf{P}\left[\max\{R_s^T, 0\} \le \frac{D}{w(s)}\sqrt{|A_s|T} + \frac{2D}{w(s)}\sqrt{2T\log\frac{|A_s|}{p}}\right] \ge 1 - p,$$

valid for every information set $s$.

Now, using the known analysis of CFR (Proposition 2), we obtain that the regret accumulated by the ESCHER iterates satisfies

$$R_i^T \le \sum_s \max\{0, R_s^T\}.$$

Hence, using a union bound over all information sets $s \in \mathcal{S}_i$ of player $i$, we find that

$$\mathbf{P}\left[R_i^T \le \left(\sum_s \frac{D\sqrt{|A_s|}}{w(s)} + \frac{2D}{w(s)}\sqrt{2\log\frac{|A_s||\mathcal{S}_i|}{p}}\right)\sqrt{T}\right] \ge 1 - p$$

$$\iff \quad \mathbf{P}\left[R_i^T \le \left(\frac{2D|\mathcal{S}_i|}{\min_s w(s)}\sqrt{2|A_s|\log\frac{|A_s||\mathcal{S}_i|}{p}}\right)\sqrt{T}\right] \ge 1 - p,$$

for all $p \in (0, 1)$. Absorbing game-dependent and sampling-policy-dependent constants yields the statements. $\qquad\square$

We remark that when the exploration policy is chosen to be the *exploration-balanced strategy* (Farina et al., 2020), then the minimum reach probability term $1/\min_s w(s)$ is upper bounded by the number of terminal states in the game, a polynomial number in the game tree. When the exploration policy is set to be the uniform strategy, $1/\min_s w(s)$ is a bounded parameter that depends on the game tree structure. In some artificial games, such as the centipede game, such a parameter is exponential in the game tree size. In most games with a reasonably balanced structure, such a parameter is polynomial in the game tree size.

### A.3   INCORPORATING FUNCTION APPROXIMATION ERRORS

We now adapt the correctness analysis above to keep into account errors in the approximation of the history value function by the deep neural network. We can model that error explicitly by modifying (4) to incorporate a history-action-dependent error $\delta(h, a)$ as follows:

$$\hat{r}_i(\pi, s, a|z) = \begin{cases} v_i(\pi, z[s]a) - v_i(\pi, z[s]) + \delta(z[s], a) & \text{if } z \in Z(s) \\ 0 & \text{otherwise.} \end{cases}$$

Repeating the analysis of Section 3, we have

$$\mathbf{E}_{z \sim \tilde{\pi}^i}[\hat{r}_i(\pi, s, a|z)] = w(s)r^c(\pi, s, a) + w(s)\sum_{h \in s}\eta_{-i}^\pi(h)\delta(h, a).$$

Propagating the error term throughout the analysis, assuming that each error term is at most $\epsilon > 0$ in magnitude, we obtain that for each infostate $s$ and $p \in (0, 1)$,

$$\mathbf{P}\left[R_s^T \le \frac{(D + \epsilon)}{w(s)}\sqrt{|A_s|T} + \epsilon + \frac{2(D + \epsilon)}{w(s)}\sqrt{2T\log\frac{|A_s|}{p}}\right] \ge 1 - p,$$

Again using the union bound across all infostates of player $i$, we obtain

$$\mathbf{P}\left[R_i^T \le \left(\frac{2(D + \epsilon)|\mathcal{S}_i|}{\min_s w(s)}\sqrt{2|A_s|\log\frac{|A_s||\mathcal{S}_i|}{p}}\right)\sqrt{T} + |\mathcal{S}_i|\epsilon\right] \ge 1 - p,$$

showing that errors in the function approximation cause an additive regret overhead linear in $\epsilon$.

## B  RELATED WORK

Superhuman performance in two-player games usually involves two components: the first focuses on finding a model-free blueprint strategy, which is the setting we focus on in this paper. The second component improves this blueprint online via model-based subgame solving and search (Burch et al., 2014; Moravcik et al., 2016; Brown et al., 2018; 2020; Brown & Sandholm, 2017b; Schmid et al., 2021). This combination of blueprint strategies with subgame solving has led to state-of the art performance in Go (Silver et al., 2017), Poker (Brown & Sandholm, 2017a; 2018; Moravčík et al., 2017), Diplomacy (Gray et al., 2020), and The Resistance: Avalon (Serrino et al., 2019). Methods that only use a blueprint have achieved state-of-the-art performance on Starcraft (Vinyals et al., 2019), Gran Turismo (Wurman et al., 2022), DouDizhu (Zha et al., 2021), Mahjohng (Li et al., 2020), and Stratego (McAleer et al., 2020; Perolat et al., 2022). Because ESCHER is a method for finding a blueprint, it can be combined with subgame solving and is complementary to these approaches. In the rest of this section we focus on other model-free methods for finding blueprints.

Deep CFR (Brown et al., 2019; Steinberger, 2019) is a general method that trains a neural network on a buffer of counterfactual values. However, Deep CFR uses external sampling, which may be impractical for games with a large branching factor, such as Stratego and Barrage Stratego. DREAM (Steinberger et al., 2020) and ARMAC (Gruslys et al., 2020) are model-free regret-based deep learning approaches. ReCFR (Liu et al., 2022) propose a bootstrap method for estimating cumulative regrets with neural networks that could potentially be combined with our method.

Neural Fictitious Self-Play (NFSP) (Heinrich & Silver, 2016) approximates fictitious play by progressively training a best response against an average of all past opponent policies using reinforcement learning. The average policy converges to an approximate Nash equilibrium in two-player zero-sum games.

Policy Space Response Oracles (PSRO) (Lanctot et al., 2017; Muller et al., 2019; Feng et al., 2021; McAleer et al., 2022) are another promising method for approximately solving very large games. PSRO maintains a population of reinforcement learning policies and iteratively trains a best response to a mixture of the opponent's population. PSRO is a fundamentally different method than the previously described methods in that in certain games it can be much faster but in other games it can take exponentially long in the worst case. Neural Extensive Form Double Oracle (NXDO) (McAleer et al., 2021) combines PSRO with extensive-form game solvers, and could potentially be combined with our method.

There is an emerging literature connecting reinforcement learning to game theory. QPG (Srinivasan et al., 2018) shows that state-conditioned $Q$-values are related to counterfactual values by a reach weighted term summed over all histories in an infostate and proposes an actor-critic algorithm that empirically converges to a NE when the learning rate is annealed. NeuRD (Hennes et al., 2020), and F-FoReL (Perolat et al., 2021) approximate replicator dynamics and follow the regularized leader, respectively, with policy gradients. Actor Critic Hedge (ACH) (Fu et al., 2022) is similar to NeuRD but uses an information set based value function. All of these policy-gradient methods do not have theory proving that they converge with high probability in extensive form games when sampling trajectories from the policy. In practice, they often perform worse than NFSP and DREAM on small games but remain promising approaches for scaling to large games (Perolat et al., 2022). Robust reinforcement learning (Morimoto & Doya, 2005; Pinto et al., 2017; Tessler et al., 2019; Lanier et al., 2022), seeks to train an RL policy to be robust against an adversarial environment. In future work we will look to apply ESCHER to this setting.

Markov games (or stochastic games) are extensive-form games where the world state information is shared among all players at each timestep, but players take simultaneous actions. Recent literature has shown that reinforcement learning algorithms converge to Nash equilibrium in two-player zero-sum Markov games (Brafman & Tennenholtz, 2002; Wei et al., 2017; Perolat et al., 2018; Xie et al., 2020; Daskalakis et al., 2020; Jin et al., 2021) and in multi-player general-sum Markov potential games (Leonardos et al., 2021; Mguni et al., 2021; Fox et al., 2022; Zhang et al., 2021; Ding et al., 2022).

## C    ADDITIONAL EXPERIMENTAL RESULTS

### C.1    DESCRIPTION OF GAME INSTANCES

We use Openspiel (Lanctot et al., 2019) for all our games. Below we list the parameters used to define each game in Openspiel.

**Leduc (`leduc_poker`)** Parameters: `{"players": 2}`
**Battleship (`battleship`)** Parameters: `{"board_width": 2, "board_height": 2,`
`        "ship_sizes": "[2]", "ship_values": "[2]", "num_shots": 3,`
`        "allow_repeated_shots": False}`
**Liar's Dice (`liars_dice`)** Parameters: `None`
**Phantom Tic Tac Toe (`phantom_ttt`)** Parameters: `None`
**Dark Hex 4 (`dark_hex`)** Parameters: `{"board_size": 4}`
**Dark Hex 5 (`dark_hex`)** Parameters: `{"board_size": 5}`
**Dark Chess (`dark_chess`)** Parameters: `None`

### C.2    TABULAR EXPERIMENTS

We compare a tabular version of ESCHER with oracle value functions to a tabular version of DREAM with oracle value functions and with OS-MCCFR. We run experiments on Leduc poker, Battleship, and Liar's dice, and use the implementations from OpenSpiel (Lanctot et al., 2019). We see in Figure 3 on the top row that ESCHER remains competitive with DREAM and OS-MCCFR on these games. On the bottom row we plot the average variance of the regret estimators over all information sets visited over an iteration window for each of these algorithms. While DREAM does improve upon OS-MCCFR, it still has orders of magnitude higher variance than ESCHER. Although this does not matter much in tabular experiments, we conjecture that high regret estimator variance makes neural network training unstable without prohibitively large buffer sizes.

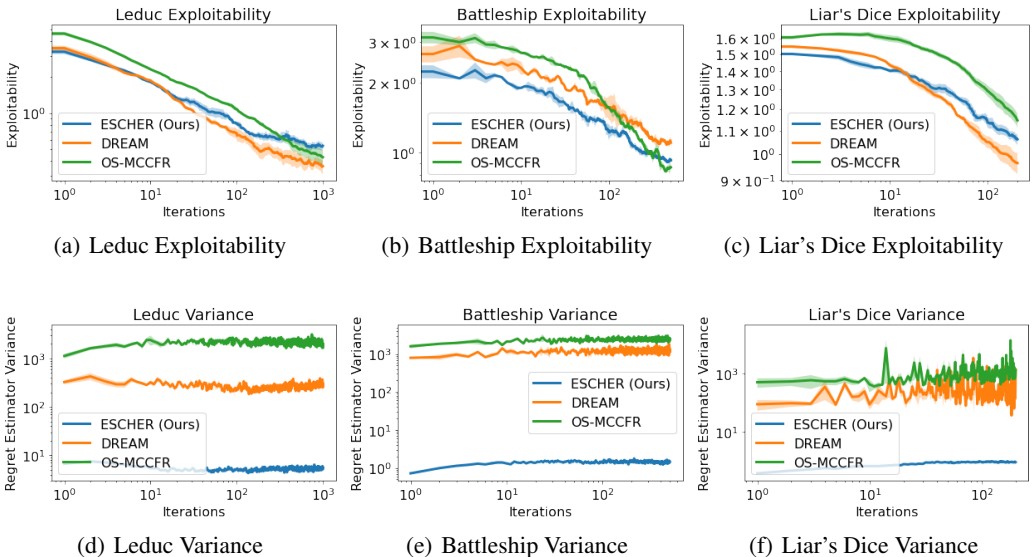

(a) Leduc Exploitability        (b) Battleship Exploitability        (c) Liar's Dice Exploitability

(d) Leduc Variance        (e) Battleship Variance        (f) Liar's Dice Variance

Figure 3: The tabular version of ESCHER with an oracle value function is competitive with the tabular version of DREAM with an oracle value function and with OS-MCCFR in terms of exploitability (top row). The regret estimator in ESCHER has orders of magnitude lower variance than those of DREAM and OS-MCCFR (bottom row).

### C.3    ADDITIONAL ABLATIONS

In this section we describe two sets of experiments. In the first set of experiments we ablate the exploration term for the sampling policy. The algorithm presented in the main paper corresponds to

exploration equal to 1, i.e. the sampling policy always plays uniform. Alternatively, we could sample from only the current policy, which we call exploration of 0. Lastly, we present results where we sample from a mixture of 0.1 times the uniform policy and 0.9 times the current policy, which we call exploration of 0.1. In the second set of experiments, we compare training the value function from scratch every iteration against not re-initializing the value function.

As shown in the top row of Figure 4, these preliminary results suggest that there is not much difference in which sampling distribution we use. However, this is likely due the the games we evaluate on. We suspect that in games such as video games, sampling from a uniform distribution will perform worse than from the current policy, because a uniform distribution will spend most of its time on bad actions. However, little is known theoretically about this on-policy setting, and it is an interesting direction for future research. On the bottom row we see that there again isn't much difference between re-initializing the value function and keeping the same value function, but keeping the same value function performs slightly better. We suspect that using best practices from learning on-policy value functions in common single-agent actor-critic algorithms will improve performance.

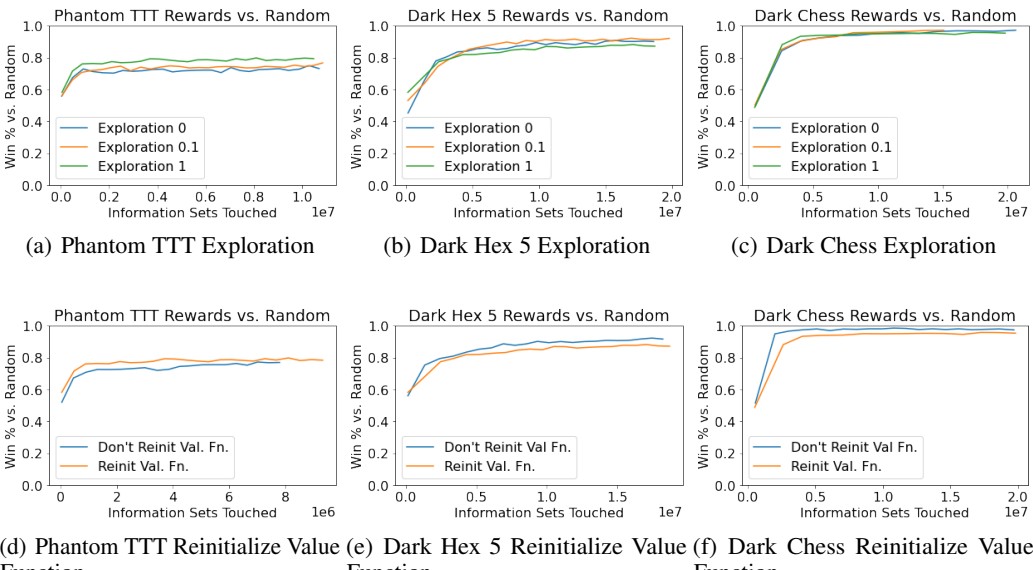

(a) Phantom TTT Exploration  (b) Dark Hex 5 Exploration  (c) Dark Chess Exploration

(d) Phantom TTT Reinitialize Value Function  (e) Dark Hex 5 Reinitialize Value Function  (f) Dark Chess Reinitialize Value Function

Figure 4: Sampling from a uniform strategy vs. sampling from the current policy or a mixture of the two does not seem to make a large difference against a random opponent (top row). Not re-initializing the value function seems to perform slightly better than training from scratch every iteration (bottom row).

# D HYPERPARAMETERS FOR DEEP EXPERIMENTS

For all deep experiments we first did a hyperparameter search that starts with good hyperparameters for flop hold 'em poker. We report the final hyperparameters used for the deep experiments. As described in the Dark Chess section, DREAM experiments on dark chess used 500 batches for advantage and average training to not run out of memory. All experiments shown include two seeds, with error bars that correspond to standard error of the mean.

## D.1 ESCHER

| Parameter | Value |
|---|---|
| n regret network traversals | 1,000 |
| n history value network traversals | 1,000 |
| batch size regret network | 2048 |
| batch size history value network | 2048 |
| train steps regret network | 5,000 |
| train steps history value network | 5,000 |
| train steps average policy network | 10,000 |

Table 3: ESCHER and Ablations Hyperparameters for Phantom TTT, Dark Hex 4, Dark Hex 5

| Parameter | Value |
|---|---|
| n regret network traversals | 1,000 |
| n history value network traversals | 1,000 |
| batch size regret network | 2048 |
| batch size history value network | 2048 |
| train steps regret network | 500 |
| train steps history value network | 500 |
| train steps average policy network | 10,000 |

Table 4: ESCHER and Ablations Hyperparameters for Variance Experiments

When computing the value function, we random noise to the current policy to induce coverage over all information sets. To do this we added $0.01$ times a uniform distribution to the current policy and renormalized.

## D.2 DREAM

We use the codebase from the original DREAM paper (Steinberger et al., 2020) with a wrapper to integrate with Openspiel (Lanctot et al., 2019) and rllib (Liang et al., 2018). When otherwise specified, we use default parameters from the DREAM codebase.

| Parameter | Value |
|---|---|
| n batches adv training | 4,000 |
| n traversals per iter | 1,000 |
| n batches per iter baseline | 1,000 |
| periodic restart | 10 |
| max n las sync simultaneously | 12 |
| mini batch size adv | 10,000 |
| max buffer size adv | 2,000,000 |
| mini batch size avrg | 10,000 |
| max buffer size avrg | 2,000,000 |
| batch size baseline | 2048 |
| n batches avrg training | 4000 |

Table 5: DREAM Hyperparameters for Phantom TTT, Dark Hex 4, Dark Hex 5

## D.3 NFSP

We use our own implementation of NFSP that uses RLLib's (Liang et al., 2018) DQN implementation and outperforms the original paper's results on Leduc poker.

| Parameter | Value |
|---|---|
| `circular buffer size` | 2e5 |
| `total rollout experience gathered each iter` | 1024 steps |
| `learning rate` | 0.01 |
| `batch size` | 4096 |
| `TD-error loss type` | MSE |
| `target network update frequency` | every 10,000 steps |
| `RL learner params` | DDQN |
| `anticipatory param` | 0.1 |
| `avg policy reservoir buffer size` | 2e6 |
| `avg policy learning starts after` | 16,000 steps |
| `avg policy learning rate` | 0.1 |
| `avg policy batch size` | 4096 |
| `avg policy optimizer` | SGD |

Table 6: NFSP Hyperparameters for Phantom TTT, Dark Hex 4, Dark Hex 5

### D.4 DARK CHESS

Hyperparameters are the same as in other deep experiments (described above), except DREAM experiments on dark chess used 500 batches for advantage and average training to not run out of memory. For these experiments only the current observation was passed in to the network for each method. As a result, we cannot expect these algorithms to learn a strong strategy on dark chess, but it is still a fair comparison. In future work we plan on doing more engineering to include more information to the networks.

## E CODE

We have open sourced our code here: `https://github.com/Sandholm-Lab/ESCHER`. Our code is built on top of the OpenSpiel (Lanctot et al., 2019) and Ray (Moritz et al., 2018) frameworks, both of which are open source and available under the Apache-2.0 license.

