# OpenReview forum: "ESCHER: Eschewing Importance Sampling in Games by Computing a History Value Function to Estimate Regret"
_ICLR.cc/2023/Conference — ICLR 2023 poster_

### Official Review · Reviewer_FHV8 · 2022-10-12

**Confidence:** 4
**Correctness:** 3
**Technical Novelty And Significance:** 3
**Empirical Novelty And Significance:** 3
**Recommendation:** 6

**Clarity, Quality, Novelty And Reproducibility:**

The paper is clear and the idea is novel. For validness I indeed have some concerns raised above.

**Strength And Weaknesses:**

The intuition looks pretty reasonable to me. However, I still have some questions regarding ESCHER before I can recommend accepting.
1. In Section 3 it's claimed sampling using a fixed distribution (what I think is the $\tilde{\pi}$ here) can avoid importance sampling. Why this is the case? I don't think this claim is supported properly.
2. In the last line of equation (6), $w(s)$ is defined via $\pi'$ but I think it comes from the sampling strategy $\tilde{\pi}$ right? Why is the inconsistency here? Is this a typo or I have missed something important?
3. It looks to me very obvious that if an oracle of value function is available, we never need to use importance sampling because we can compute the counterfactual regret directly. I don't think this makes any essential improvement, since we may still need important sampling to estimate these value functions from samples. How should this be handled? Actually, I think this is more important than the content discussed in detail in section 3, and the authors may want to explain more about that.
4. I think it will be super helpful it the authors can give some intuition on why it's possible to avoid using importance sampling at all. Say for the strategic-form game, can we also avoid using importance sampling in Mento Carlo using the proposed approach? Or is there any certain character of  EFG such makes this possible?

**Summary Of The Paper:**

The paper develops an algorithm, ESCHER, to reduce the variance of CFR with deep learning by avoiding importance sampling.

**Summary Of The Review:**

The paper develops an algorithm, ESCHER, to reduce the variance of CFR with deep learning by avoiding importance sampling. The motivation looks sound to me but I need to be convinced how the remaining part hidden from oracle of value function can be addressed before recommending accept.

---

> ### Author Response · Authors · 2022-11-14
> **Author Response**
>
> Thank you for your helpful comments.
>
> 1. The assumption of a fixed sampling distribution is used to prove Theorem 1., which shows that using the estimator in equation 5, which doesn’t use importance sampling, is sound. We have updated the wording to make this more clear.
> 2. That was indeed a typo and we have fixed it.
> 3. We estimate the value function without using importance sampling by simply sampling on-policy trajectories and regressing on future reward at a given history. This approach is common in deep reinforcement learning (e.g. PPO and A2C). We have updated the paper to better describe how we train the value function.
> 4. Yes, if we have an oracle value function in a strategic-form game, this is equivalent to the full-feedback setting where regret matching is able to be no regret without importance sampling. We have included a section describing this case (including a standalone proof of its regret bound that might be otherwise hard to track down exactly in the rather fragmented literature) in section 7.1.

---

> > ### Comment · Reviewer_FHV8 · 2022-11-14
> > **Clarifications**
> >
> > Just want to make sure I'm understanding your response correctly:
> > 1. the estimator in equation 5 of course avoids using importance sampling, but I think the reason is we assume an oracle of value function is available here. How is this related to the fixes sampling distribution?
> > 3. can you point out where is the estimation of the value function explained? I have a look at the new version but didn't find it.
> > 4. I agree the oracle is equivalent to full info feedback. The problem is: when full info feedback is not available, how can we still approximate such an oracle without importance sampling? I think this lies at the heart of the argument but I don't think it's described in detail in the main text. (please correct me if you actually have done that)

---

> > > ### Author Response · Authors · 2022-11-15
> > > **Response to Clarifications**
> > >
> > > 1. If we understand the question correctly, the reviewer is asking whether the ability to avoid the importance sampling term is fundamentally a consequence of the fact that in this paper we assume oracle access to a value function—which is a stronger assumption than what is done, say, in regular MCCFR. This is an interesting question and, as far as we know, the answer is yes: it is not known that importance sampling can be avoided with the weaker oracle, that is, without assuming access to a history value function.
> > >
> > >     However, we stress that we prefer access to a value function also due to practical considerations: in large games, it is easy to train such an oracle, and it would make little sense to restrict ourselves to a weaker oracle. So, even if we had a way of eschewing importance sampling using a weaker oracle, we believe such an algorithm would likely be dominated by ESCHER.
> > >
> > >     Finally, we note that our analysis accounts for the possibility that the (e.g., neural-network-based) oracle value function has some bounded error, and predicts the degradation in performance of ESCHER as a function of that error’s magnitude.
> > >
> > > 2. We included new text in blue on page 8 describing how we train the value function. We also have uploaded our code. Please let us know if you think we should add some details in particular about the value function to the appendix.
> > >
> > > 3. If the current policy takes all actions with some probability, then the value function of the current policy can be estimated by training a neural network to estimate future returns from on-policy trajectories conditioned on histories. These on-policy trajectories are collected by rolling out the current policy for both players. The future return targets are not modified with importance sampling. This is equivalent to Monte-Carlo estimation in single-agent RL (e.g. Sutton and Barto chapter 5) and is widely used in deep actor-critic implementations. Since the regret-matching policy does not take all actions with equal probability, we add a small amount of noise to the policy.

---

> > > > ### Comment · Reviewer_FHV8 · 2022-11-15
> > > > **Response**
> > > >
> > > > Thanks for your prompt update.
> > > >
> > > > Now I think I understand: the reason why we no longer need importance sampling is essentially we use the roll out trajectory (and the corresponding reward) to train the value function estimator. This is more like what people do in policy gradient methods like REINFORCE.
> > > >
> > > > The picture now looks much more reasonable to me. However my concern now is mainly about novelty:
> > > > 1. When given the value function oracle, I think we can apply some non-Mento Carlo methods like CFR directly, which can give a similar guarantee to Thm. 1. Then what's the point of including section 3?
> > > > 2. I think ESCHER should mostly be compared to methods like deep CFR, as the authors already do in Section 4. The main difference is now (8) is used. I agree here is some methodological novelty but then the most important part would be the experiments.
> > > >
> > > > Regarding experiments I indeed have some questions:
> > > > 1. I think usually people measure the performance by looking at how far it converges to Nash Equilibirum (at least in 2p0s games). Why here we play with uniform randomly agents or agents taking a baseline algorithm? I mean, even if ESCHER is performing better it not necessarily support the claim that it minimizes the regret or find the NE faster.
> > > > 2. The improvement is only significant when playing with random agents. This doesn't sound like a significant improvement to me because we have a lot of different ways to exploit uniformly random agents. The key here is how good can we minimize regret (as in Thm. 1). Of course I agree regret is some times not very easily to measure. But then one can look at the output policy and its exploitability.

---

> > > > > ### Author Response · Authors · 2022-11-16
> > > > > **Author Response**
> > > > >
> > > > > Thanks for engaging with us, and we are glad to hear our responses have been useful and the picture is looking more reasonable to you! We are hoping with this round (and possibly future ones too) we will be able to resolve the remaining reservations about novelty.
> > > > >
> > > > >
> > > > > Re “This is more like what people do in policy gradient methods like REINFORCE.”
> > > > >
> > > > > You are correct in that we use the rollout trajectory and the corresponding reward to train the value function. However, the way we train the value function is different from methods like REINFORCE, which do **policy** improvement by taking a policy gradient. Instead, we train the **value** function using the simpler method of regression on Monte Carlo estimates of the reward (for policy evaluation).
> > > > >
> > > > >
> > > > > Re “1. [..] I think we can apply some non-Mento Carlo methods like CFR directly”
> > > > >
> > > > > We think it is crucial to keep in mind that in this paper we are interested in games that are far too large to fit into memory. Non-Monte Carlo methods such as CFR require that all probabilities of all actions be updated at each iteration, which is not possible at the scale we are interested in. As a result, we cannot do anything else but assume access to *trajectories*. So comparisons to Deep CFR and DREAM are more relevant. Section 3  should be regarded as a mere description of an idealized version of ESCHER to provide theoretical grounding—it should not be viewed as a standalone algorithm. (We explain this caveat in the first paragraph of section 3.)
> > > > >
> > > > > Re “the main difference is (..) is now used”
> > > > > This is correct: the main difference is now (7) is used as the regret estimator instead of the importance sampling estimator of DREAM and Deep CFR. This change is conceptually atomic, but makes all the difference. When looking at the big pictures, it allows us to scale to large games while DREAM does not.
> > > > >
> > > > > Re “how far it converges to Nash Equilibrium”
> > > > > We totally agree: it is a common desideratum of this line of work to demonstrate (or at least quantify) convergence to an approximate Nash equilibrium. The common reality for these large games, unfortunately, is that computing the exploitability of the computed solution is infeasible because the game is too large. The alternative could be to train an exploitability estimator, but this leads to a circular issue: how can we certify that the exploitability estimate is accurate if we do not know the equilibrium value and have no way of computing best responses?
> > > > > Including tabular experiments on the largest games for which we can compute exploitability and show that ESCHER does indeed converge to an approximate Nash equilibrium was an attempt to ameliorate the issue and instill confidence in our experimental setup. These results are included in the appendix. In large games, since we cannot compute exploitability, the common practice is to evaluate against other algorithms or humans (e.g. DeepNash for Stratego, Libratus for poker, AlphaGo for Go, and AlphaStar for StarCraft).
> > > > >
> > > > > At any rate, we note that even without knowing the distance from equilibrium, we believe our experiments paint a pretty compelling picture of how much better ESCHER is over the other methods. At a minimum, the fact that those other methods seem unable to even learn to beat uniformly random agents remotely as well as ESCHER should be convincing evidence of the improvement that our paper brings to the table.
> > > > >
> > > > > Re “ But then one can look at the output policy and its exploitability.”
> > > > > We hope our previous response convinced the reviewer that what they propose is not a possibility in large games.
> > > > >
> > > > > Re “The improvement is only significant when playing with random agents.”
> > > > > This might be a misunderstanding in what the plots show on the axes. In the second row of plots in Figure 1 and Figure 2 we show two lines. These are  **head-to-head performance of ESCHER vs DREAM and ESCHER vs NFSP**. In both cases, we find that ESCHER significantly outperforms both DREAM and NFSP head to head in Dark Hex 5 and Dark Chess, but roughly tie in Phantom Tic Tac Toe. The fact that the lines are close to each other means that ESCHER outperforms both algorithms by similar margins, and **not** that ESCHER leads to a similar expected utility as DREAM/NFSP. We further note that the performance difference between ESCHER and the other methods becomes larger as the game size increases.

---

> > > > > > ### Comment · Reviewer_FHV8 · 2022-11-16
> > > > > > **Thanks for handling my concerns**
> > > > > >
> > > > > > I have adjusted my score accordingly. There are still some writing issues that I hope the authors can improve upon. (listed below)
> > > > > >
> > > > > > Re “1. [..] I think we can apply some non-Mento Carlo methods like CFR directly”
> > > > > >
> > > > > > I think here I am more talking about the writing. In Section 3 it's about the tabular case, right? My point is this part doesn't seem very important to me and the guarantee (regret bound) is not directly related to the experiment results (competing with different learning algs). I think you may want to compress this section and add more discussion for section 4.
> > > > > >
> > > > > > Re “the main difference is (..) is now used”
> > > > > >
> > > > > > If this is crucial, I think you may want to stress this instead of "avoiding importance sampling", since this is the main novelty that makes the difference.

---

> > > > > > > ### Author Response · Authors · 2022-11-18
> > > > > > > **We have re-written sections 3 and 4**
> > > > > > >
> > > > > > > Thank you for this helpful comment. We have rewritten sections 3 and 4 and have merged the two into one section. In this section, we introduce ESCHER as the deep version, and only talk about the tabular version in the theoretical results subsection. We have also highlighted the main difference being the different regret estimator. We believe that after implementing your suggestion the paper is written much better and is less confusing.
> > > > > > >
> > > > > > > Would you please be able to take a look at the new section and let us know if you have any further suggestions? If not, would you please consider revising your score?

---

### Official Review · Reviewer_yTxH · 2022-10-24

**Confidence:** 3
**Correctness:** 4
**Technical Novelty And Significance:** 3
**Empirical Novelty And Significance:** 3
**Recommendation:** 6

**Clarity, Quality, Novelty And Reproducibility:**

#Review

Dealing with the high variance of MCCFR type algorithms, especially in the non-tabular case is an import question. Thus, I think  the proposed solution is a valuable contribution since it seems to solve this issue. However, the presentation of the submission does not meet the requirement of what I would consider acceptable for this venue. In particular several key quantities are ill-defined or  not defined at all, see specific comments. Also, some parts of the algorithms that should be discuss thoroughly are left unspecified, see specific comments. For these reasons, in the overall, it is hard to asses the quality of the presented method. Furthermore, we could expect more discussion (experimentally and theoretically trough min_s w(s)) on the choice of the sampling policy and the effects of this choice.  The proofs seem correct as far as I checked. We would expect access to the code used for the experiments.



#General comments:
- The code for the experiments is not available.
- As notice in Figure 3, it seems that the high variance is not an issue in the tabular case. Can you elaborate on the conjecture 'that high regret estimator variance makes neural network training unstable without prohibitively large buffer sizes'. Could you also think about experiments that may support this conjecture.

#Specific comments:
- P2, Section 2: Could the utility be non-zero at non terminal state?
- P2, Section 2: Why the observation should depends on (w,a,w') and not only w'?
- P2, Section 2: Abbreviation RL not defined.
- P2, Section 2: The best response is not necessarily unique then BR could be a set as you defined it.
- P3, Section 2.1: Section 8.
- P3, Section 2.1: Could you define precisely \eta^\pi(h)? In (1), \eta^\pi(h,z) is not defined neither u_i(z) (= \mathcal{U}_i(z)?) . Similarly (2) is hard to read without the definition of the used quantities.
- P3, Section 2.1: The average policy (of CFR?) converges in which sense? What is e(\bar{\pi}^T)?
- P5, above (4): Can you explain why sampling uniformly over actions allows to roughly 'visits every information set equally'?
- P5, (5): It seems that q(\pi,h,a) is never properly defined.
- P5, below (6): Policy b instead of \pi'?
- P6, Algorithm 1: For self-completeness you could defined what is the regret matching algorithm. And can you explain how exactly you can compute \hat{r} from the sampled trajectory \tau from \tilde{p}. In particular is it easy to compute q_i and v_i? In general it could be useful to state to which oracles you agent has access.
- P6, Theorem 1: What do you mean by 'the regret accumulated by each agent', is this regret defined somewhere?
- P6, Section 4: Which policies do you use exactly to fill the first replay buffer?
- P7, Table 2: Add a reference for the baselines you compare with. And bootstrapped baseline in the first row.
- P7, Section 4: What is the quantity that you want to approximate with R_i(s,a|\psi) exactly? Is there any difficulties in approximating a running sum? How do you concretely handle the dependence over the policy \pi in the q-value network  q_i (\pi, h, a|\theta)? How do you train the average policy network and the value network exactly?
- P6, Algorithm 2: It seems that the replay buffer and update of the value network do not appears in the algorithm.
- P8, Figure 1: Can you precise how many seeds did you use in the experiments?
- P13, Proposition 1: \hat{\pi}_i and v_i(\pi) is not defined.
- P14: Can you provide a pointer for the regret bound of regret matching or even rewrite the results.
- P15: I'm not sure I understand the end of the sentence: 'translate linearly into additive regret overhead'.
- P15, Section 7: Can you provide an upper bound on 1/min_s w(s) in the case where you use as sampling policy the one uniform over actions.

**Strength And Weaknesses:**

*Strength

- An importance sampling free algorithm for imperfect information games

*Weaknesses

- Lack of clarity.

**Summary Of The Paper:**

The authors study the approximation of  Nash equilibrium in very large extensive-form games with prefect recall. One line of research approximates with the use of neural networks the tabular algorithm: counterfactual regret minimization (CFR). One issue with these methods is that they train a neural network with targets that can have extremely high variance due to the use of importance sampling estimation (of the counterfactual regret). The authors propose the ESCHER algorithm a model-free method that does not require any importance sampling. They prove that the tabular version is guaranteed to converge to an approximate Nash equilibrium with high probability. Experimentally they show that both the tabular version and the deep version of ESCHER enjoy a smaller variance than several baselines such that Outcome Sampling MCCFR in the tabular case and NFSP and DREAM in the function-approximation case. Furthermore they show that ESCHER outperform NFSP and DREAM on three games: Phantom Tic-Tac-Toe, Dark Hex 5 and Dark Chess.

**Summary Of The Review:**

See above.

---

> ### Author Response · Authors · 2022-11-10
> **Author Response**
>
> Thank you for your detailed and helpful comments.
>
> We have uploaded a zip file with the code. We have included experiments in the appendix examining the effect of different sampling distributions and find that there is not a large difference between our approach and an on-policy approach. However, we currently do not theoretically understand the on-policy versions of ESCHER and we propose investigating them as future work.
>
> Yes, the high variance is not an issue in the tabular case. The fact that the tabular versions of DREAM and Deep CFR work in the tabular case but not in the neural case provides evidence that the neural networks are not approximating the tabular versions of these algorithms. Since DREAM and Deep CFR work well in games with low target regret variance like poker, but poorly in games with high variance, we believe that a likely reason the neural networks aren’t approximating the tabular version of the algorithm is due to the variance. In support of this, we further note that the variance increases going from Phantom TTT to Dark Hex 5 to Dark Chess, and the performance of DREAM indeed gets worse as the variance increases, while the performance of ESCHER remains good (and progressively better than DREAM) as expected. We have updated the paper to better explain and discuss this topic.
>
> Thank you for the list of specific comments. We agree with every suggestion and have updated the paper to reflect all of these changes. In particular, we have added two pages of detailed background on regret matching (including a standalone proof of its regret bound that might be otherwise hard to track down exactly in the rather fragmented literature) in the appendix.
>
> Thank you again for the comments, they have made the paper much better. Please let us know if you have any additional comments or questions we can address.

---

> > ### Author Response · Authors · 2022-11-16
> > **Any Further Questions?**
> >
> > Are there any remaining changes you would like us to make or questions you would like us to answer? If you don't have any remaining questions or requests, could you please update your score to reflect your updated impression of the paper? Thanks!

---

> > > ### Comment · Reviewer_yTxH · 2022-12-09
> > > **Post rebuttal**
> > >
> > > Thanks for the clarifications and updates of the submission. I have read the authors' rebuttal and other reviews.
> > > The authors answered to my questions or update the submission accordingly. Therefore, I update my score to 6.
> > >
> > > As a final comment could you precise in Alg. 1 how the policy \pi is updated as in Alg. 2. In particular I'm a bit confused with the on policy sampling, P5 "the on-policy sampling is performed [...] uniformly-random policy", maybe you can precise this in (5).

---

### Official Review · Reviewer_VgYx · 2022-10-25

**Confidence:** 3
**Clarity, Quality, Novelty And Reproducibility:** sound
**Correctness:** 3
**Technical Novelty And Significance:** 3
**Empirical Novelty And Significance:** 3
**Recommendation:** 6

**Strength And Weaknesses:**

Strength:

(1) This paper is well-written and easy to follow, the background part and related work are very detailed.

(2) The authors give proof of the upper bound of ESCHER assuming a fixed sampling policy.

(3) The additional Table 2 in this version clearly shows the differences between ESCHER and previous works

Weaknesses:

(1) ESCHER proposes a two-stage iteration, which may not be a good practice (also discussed in the Limitation part). The gap between the outer fixed sampling policy and the inner regret learning should be addressed experimentally or theoretically.

(2) In the neurips2022 rebuttal, the authors also agreed that "Yes, retraining a value function every iteration is slow, but that's not the main point of the paper", therefore, it is expected to estimate the additional overheads for training the value function. And some of the results should be presented in another way. For example, the x-axis in Figure 3 is the number of iterations, which may guide the reader that ESCHER converges much faster than DREAM and OS-MCCFR, but the effort within an iteration should be noted.

(3) The correlation between the high variance of the regret estimator and the low performance/winrate of the algorithm should be addressed. Since the algorithm may have a larger variance due to its unique settings and hyperparameters but can perform well in practice (e.g., proximal policy optimization), the authors are encouraged to give more evidence why ESCHER is better.





**Summary Of The Paper:**

This paper proposes to learn a history-dependent value function and sample actions from a fixed sampling policy to replace the importance sampling and reduce variance.

Overall, this paper is well-written and easy to follow, the reviewer is not very professional in the field of CFR, but still catches most of the content.

*** The reviewer has reviewed this paper in neurips2022, since most of the concerns are addressed by the reply at that time, the reviewer would suggest acceptance at this time.

There are still some minor questions as follows.

**Summary Of The Review:**

see above

---

> ### Author Response · Authors · 2022-11-14
> **Author Response**
>
> Thank you for your positive and helpful comments.
>
> 1. We have run experiments where we sample from the current policy instead of a fixed policy. By sampling on-policy, we can remove the need for two separate iterations, and can ideally update faster by not continuing to sample parts of the game tree that are known to be bad. However, we currently do not theoretically understand the on-policy versions of ESCHER—we propose investigating them as future work. We have included the new results in the appendix where we compare sampling from the current policy, a uniform policy (as in the main paper), and a mixture of the two. We find that there is not a large difference in performance.
> 2. We have also included new results in the appendix where we do not reinitialize the value function every iteration, and show that not reinitializing the value function performs slightly better. In future work we will investigate using common best practices for estimating on-policy value functions from deep reinforcement learning.
> 3. The fact that the tabular versions of DREAM and Deep CFR work in the tabular case but not in the neural case provides evidence that the neural networks are not approximating the tabular versions of these algorithms. Since DREAM and Deep CFR work well in games where the variance of the estimated regret is low, like poker, but poorly in games with high variance, like Dark Chess, we believe that a likely reason the neural networks aren’t approximating the tabular version of the algorithm is due to the variance. Also note that the variance increases going from Phantom TTT to Dark Hex 5 to Dark Chess, and the performance of DREAM gets worse as the variance increases. Finally, we note that ESCHER has very low variance and performs very well compared to DREAM. While these observations do not ultimately prove that the reason why ESCHER performs better than DREAM and Deep CFR is due to variance, we believe they provide strong evidence in support of that hypothesis. We have updated the paper to better explain and discuss this topic.

---

> > ### Author Response · Authors · 2022-11-16
> > **Any Further Questions?**
> >
> > Are there any remaining changes you would like us to make or questions you would like us to answer? If you don't have any remaining questions or requests, could you please update your score to reflect your updated impression of the paper? Thanks!

---

> > > ### Comment · Reviewer_VgYx · 2022-11-21
> > > **Response**
> > >
> > > Thanks for the response, since the reviewer is not very familiar with this field, the opinion of the paper has not changed in light of the authors' response.
> > > Therefore I will keep my current rating as a weak acceptance.

---

### Author Response · Authors · 2022-12-09
**A Request to the AC to Prompt Discussion**

We believe we have addressed all of the reviewers' feedback but we have not heard back from two reviewers. In particular, reviewer yTxH has not responded to our first comment where we address all their concerns. Could you please ask the reviewers to post comments and update their score if we have addressed their concerns?

---

### Decision · Program_Chairs · 2023-01-20

**Decision:**

Accept: poster

**Justification For Why Not Higher Score:**

It would be stronger if the authors could explain why the uniform policy works for these experiments and what is the bound of the const in Theorem 1.


**Justification For Why Not Lower Score:**

The experiments demonstrate the efficacy of the proposed approach and the paper can be accepted.

**Metareview: Summary, Strengths And Weaknesses:**

Summary:
The paper develops an algorithm, ESCHER, to reduce the variance of CFR with deep learning by avoiding importance sampling in games. . They prove a theory for the tabular setting that the algorithm is guaranteed to converge to an approximate Nash equilibrium with high probability. Experimentally, they show that ESCHER enjoy a smaller variance than several baselines. Furthermore they show that ESCHER outperform NFSP and DREAM on three games: Phantom Tic-Tac-Toe, Dark Hex 5 and Dark Chess.

Strength:
- Paper well-written and easy to follow
- An importance sampling free algorithm for imperfect information games
- Algorithm outperforms baselines in a number of games

Weakness:
- The biggest weakness of the ESCHER approach is the sampling policy. It appears that the policy is an arbitrary or uniform policy. It is hard to believe such a policy can work in any environment that need to make sequential decisions. The theory part requires this assumption to provide guarantees. But the constant hiding in Theorem 1 can be exponential. Indeed, experimentally, the algorithm has good performance. It would be stronger if the authors could explain why the uniform policy works for these cases and what is the bound of the const in Theorem 1.


**Note From Pc:**

if the above contains the word "oral" or "spotlight" please see: "oral" presentation means -> notable-top-5% and "spotlight" means -> notable-top-25%. As stated in our emails, we are disassociating presentation type from AC recommendations

**Summary Of Ac-Reviewer Meeting:**

The paper is on the borderline. The AC consulted the reviewers, who believes that theory part is not convincing. The experiments demonstrate the efficacy of the proposed approach and the paper can be accepted.